# Proteomic Analysis of Tear Films in Healthy Female and Male Dogs Using MALDI-TOF (Matrix Assisted Laser Desortion/Ionization Time-of-Flight) Mass Spectrometry

**DOI:** 10.3390/ani15070904

**Published:** 2025-03-21

**Authors:** Dagmara Winiarczyk, Mateusz Winiarczyk, Katarzyna Michalak

**Affiliations:** 1Department of Internal Diseases of Small Animals, University of Life Sciences of Lublin, 20-400 Lublin, Poland; 2Department of Vitreoretinal Surgery, Medical University of Lublin, 20-059 Lublin, Poland; winiarm86@gmail.com; 3Department of Epizootiology, University of Life Sciences of Lublin, 20-400 Lublin, Poland; kat.michalak@gmail.com

**Keywords:** proteomics, tear film, healthy dog, animal model, ocular pathology

## Abstract

The study investigates sex-based differences in the tear film proteomes of healthy male and female dogs using Matrix-Assisted Laser Desorption/Ionization Time-of-Flight mass spectrometry (MALDI-TOF). The objective was to identify protein variations that could explain physiological differences and susceptibility to ocular diseases. Tear samples were collected from 22 dogs, and proteomic analysis revealed 446 common protein spots, with 8 exhibiting statistically significant differences between sexes. Seven proteins, including TIMP-2, PFK, and Annexin A13, were upregulated in females, while IL-33 was higher in males, indicating potential hormonal influences on tear film composition. These proteins are associated with tissue protection, metabolism, and immune responses, indicating potential hormonal influences on ocular surface homeostasis. The findings contribute to the understanding of tear film composition and may serve as a basis for further research into tear film biomarkers. This knowledge could support the development of non-invasive diagnostic approaches in veterinary ophthalmology.

## 1. Introduction

The tear film is a complex and dynamic multilayered structure that covers the ocular surface consisting of three layers: the outermost lipid layer, the middle aqueous layer and the innermost mucin layer. This structure maintains the integrity, lubrication and optical clarity of the ocular surface. The lipid layer, composed mainly of polar and non-polar lipids, acts as a barrier to prevent evaporation of the underlying aqueous layer. The aqueous layer, composed of electrolytes, proteins and enzymes, nourishes and protects the corneal and conjunctival epithelium. The mucin layer, produced by goblet cells in the conjunctiva, facilitates the spreading and adherence of tears on the ocular surface, ensuring uniform coverage and stability. The dysfunction of any layer of the tear film can lead to ocular surface disorders such as dry eye syndrome, characterized by symptoms of ocular discomfort and visual disturbance.

Tear film proteomics involves the comprehensive analysis of the protein composition of this complex biological fluid. Proteomic analysis has become an important tool in biomedical and veterinary research [1,2,3,4,5,6,7,8]. This approach utilizes techniques such as mass spectrometry and chromatography to identify and quantify proteins present in tears. The tear film proteomes include a wide range of proteins, including enzymes, antibodies, cytokines, growth factors and structural proteins [9].

In human ophthalmology, tear proteomic analysis has been extensively explored for its potential applications in disease diagnostics and biomarker discovery. Studies have identified proteomic signatures associated with conditions such as dry eye disease, keratoconus, diabetic retinopathy and neurodegenerative disorders, including Alzheimer’s disease [10,11,12,13,14]. While proteomic analyses in human tear films remain largely focused on basic research, some potential clinical applications are emerging, particularly in early disease detection [15]. However, the translation of tear proteomics into routine clinical practice is still in its early stages, and further validation studies are required to establish standardized protocols and diagnostic thresholds.

Research in veterinary medicine regarding the proteomics of body fluid has been ongoing in several studies; however, there are few publications in the literature in the field of protein profiling the tear films of animals, especially dogs [1,4,16,17,18,19].

In a previous study by our group, we identified 125 proteins in the tear films of healthy dogs [4], and in another study, we identified 9 differentiating tear proteins in dogs with diabetes [20] and 5 significantly differentially expressed proteins in dogs with diabetes with retinopathy [18].

The aim of this study was to perform a comparative analysis of the tear film proteome in a healthy female and male dogs. This approach may facilitate the development of a non-invasive method to detect potential indicators in tear films.

To date, there are no scientific data on the comparative analysis of tear films in healthy dogs which take sex differences into account.

## 2. Materials and Methods

### 2.1. Study Design and Sample Collection

Tear samples were collected from 22 healthy dogs (11 females and 11 males) using special standard Schirmer strips. All dogs were sexually intact, and the time of tear film sample collection in female dogs corresponded to the anestrus phase. During tear film collection, male dogs had no contact with female dogs. The study included dogs of different breeds (3 Dachshunds, 4 Pugs, 4 Poodles, 2 Cavalier Charles Spaniels, 2 Beagles and 7 mixed breeds). The dogs were aged between 4 and 9 years (median 7.1 years). The animals were recruited during routine veterinary visits at the Faculty of Veterinary Medicine, University of Life Sciences, in Lublin. Informed consent was obtained from the owners prior to clinical examination and sampling. Each animal underwent a clinical examination, including blood serum biochemistry, hematology and urinalysis, and a comprehensive ophthalmic examination, including the evaluation of the anterior segment and fundus and the measurement of intraocular pressure. All animals included in the study showed no signs of disease.

As the samples were obtained during standard veterinary diagnostic procedures, ethical approval by the Local Committee for Ethics in Animal Experimentation was not required under Polish law. The study was conducted in accordance with European Union Directive 2010/63/EU and followed ARRIVE guidelines.

The dogs were divided into two groups according to sex, with 11 females in the female group (FG) and 11 males in the male group (MG).

### 2.2. Tear Sample Preparation

Tear film samples were collected using Schirmer strips (TearFlo, HUB Pharmaceuticals, North Liberty, IA, USA), which were placed in the inferior conjunctival sac of both eyes at approximately one-third of the distance from the medial to the lateral canthus. The collection procedure was performed without anesthesia, following previously established methods. All samples were collected by the same investigator (DW) between 7:00 and 9:00 a.m., using sterile gloves to minimize variability. Since no standardized protocol for tear film collection in proteomic studies currently exists, a uniform methodology was applied. The Schirmer strips remained in place for 5 min before being removed. Immediately after removal, the strips were transferred to 1.5 mL Eppendorf tubes and frozen at −80 °C without buffer. Protein extraction was performed in a urea buffer for 3 h at 4 °C, with the addition of a protease inhibitor cocktail (Sigma, P8340, Saint Louis, MI, USA). After the extraction process, the Schirmer strips were removed, and the samples were centrifuged at 5000 rpm for 10 min at 4 °C. The resulting supernatants were collected and stored at −80 °C for further analysis.

### 2.3. Protein Cleaning and Precipitation

The concentration of proteins was determined by measuring the absorbance at 280 nm (MaestroNano Spectrophotometer, Maestrogen, Hsinchu City 30091, Taiwan). Tear fluid containing 150 µg of protein was placed in a 1.5 mL test tube and water was added until the mixture reached 100 µL. The ReadyPrep 2-D cleanup kit (Bio-Rad, Warszawa, Poland) was used to improve the electrophoresis results and for quantitative protein precipitation. Finally, the obtained protein pellets were dissolved in 300 µL of rehydration buffer (ReadyPrep 2-D Starter Kit Rehydration, Bio-Rad, Warszawa, Poland). The resulting solutions were then applied directly to IPG strips (17 cm, pH 3–10, Bio-Rad, Warszawa, Poland) for 12 h in-gel rehydration.

### 2.4. Electrophoresis and Protein Spot Analysis

Isoelectric focusing was performed using a Hoefer IEF100 system, applying a total of 60 kVh with a maximum current of 50 µA per strip. Following isoelectric focusing, the proteins were subjected to two equilibration steps. The first equilibration was performed in a buffer containing 2% dithiothreitol (DTT), while the second step utilized a buffer containing 2.5% iodoacetamide (IAA). Each equilibration step lasted 15 min. The second-dimension electrophoresis was carried out using 12.5% polyacrylamide gels in a PROTEAN II xi Cell system (Bio-Rad) with Tris-Glycine running buffer. The gels were subsequently stained using a silver-staining protocol according to Shevchenko et al. and digitized using a GE Healthcare imager (Poznan, Poland).

Protein expression analysis was performed using Delta2D software version 4.8 (DECODON, Greifswald, Germany), which allowed for the quantification of protein spots and the creation of expression profiles. The software employed a warping algorithm to align gel images, generating a fused proteome map that included all protein spots identified throughout the experiment. Manual verification was performed to exclude false positives and false negatives. The ratio quotient for the female group’s means of relative spot volumes was presented: the volume of a given spot in the male group was the denominator of the ratio parameter. The ratio quotient (Rt) value for a given expressed protein was used >1.5 as the up-regulation threshold and <0.67 as the downregulation threshold [21,22,23].

### 2.5. Protein Identification via MALDI TOF

Protein spots selected for identification were excised from the gels, placed in microtubes, rinsed with water, and destained. Reduction and alkylation were performed using dithiothreitol and iodoacetamide solutions. Trypsin digestion was conducted by covering the gel fragments with trypsin solution, followed by the addition of 50 mM ammonium bicarbonate solution. The prepared tubes were incubated at 37 °C for at least 8 h. After digestion, peptides were extracted using a three-step protocol in an ultrasonic bath with an extraction solution composed of acetonitrile, water, and trifluoroacetic acid in a volumetric ratio of 50:45:5. The extracted peptides were concentrated using a CentriVap system (Labconco, Kansas City, MI, USA) and purified with ZipTip C18 columns (Merck, Darmstadt, Germany).

The peptide extracts were applied to an AnchorChip MALDI plate with a hydrophobic coating and calibrator anchors. The dried peptide spots were covered with an equal volume of α-cyano-4-hydroxycinnamic acid (HCCA) matrix (Bruker, Bremen, Germany). Mass spectra were acquired using an Ultraflex III MALDI TOF/TOF spectrometer (Bruker) in the 700–4000 Da *m*/*z* range. Spectral processing was performed with flexAnalysis 3.0 software (Bruker). After contaminant removal, peak lists were transferred to BioTools 3.2 (Bruker) and compared with the Swiss-Prot database using Mascot 2.2 (Matrix Science, London, UK). The taxonomy was set to “Bony Vertebrates”, and the maximum mass error did not exceed 0.3 Da. Proteins were considered statistically significant if their Mascot score exceeded 56. For proteins with lower scores, further analysis was conducted in MS/MS tandem mode.

### 2.6. Statistical and Gene Ontology Analysis

All statistical analyses were conducted separately from the protein processing steps to improve clarity. Normalized protein spot volumes were compared between groups using a one-way ANOVA test. A *p*-value of less than 0.05 was considered statistically significant.

For Gene Ontology (GO), the Panther program PANTHER (Protein Analysis Through Evolutionary Relationships) classification system (http://www.pantherdb.org accessed on 18 March 2024) was used to assign identified proteins to the appropriate biological process. This tool enabled the categorization of proteins based on their molecular functions and involvement in specific biological pathways.

## 3. Results

Our study revealed 446 common tear film protein spots from the healthy male and female groups. Of these, eight proteins showed statistically significant differential expression (*p* ≤ 0.05) and were excised from the electrophoretic gel and positively identified by MALDI-TOF MS. Table 1 reveals the list of positively identified proteins with their names, genes and UniProt base accession numbers. Using the Delta2D program, seven of the eight proteins were classified as up-regulated in the female group and one of the eight proteins was classified as down-regulated (Table 1, Figure 1). Figure 1 reveals representative two-dimensional electrophoresis gel spots of significantly (*p* ≤ 0.05) differentially expressed proteins in the female group compared to the male group. Figure 2 reveals a merged image of the condensed electrophoretic gels from the entire experiment.

The gene ontology analysis revealed that the identified tear film proteins are associated with diverse cellular processes (Figure 3), including immune response (interleukin-33), metabolic regulation (protein phosphatase 1 regulatory subunit 1A), intracellular transport (Ras-related protein Rab-21), cell signaling (dual serine/threonine and tyrosine protein kinase), and tissue protection (metalloproteinase inhibitor 2).

## 4. Discussion

Despite the well-developed veterinary ophthalmology of the dog, the literature in the field of molecular studies of tear films is sparse and the in-depth analysis of the protein composition of the normal tear film is lacking. Most of the information on the protein profile comes from the time when the accuracy of the results obtained was largely limited by the analytical methods used. In this context, a systematic study using the latest achievements in proteomic technology should start from the analysis of the normal tear film protein profile in healthy subjects. This manuscript is an introduction to population studies to determine the physiological levels of important proteins in the tear films of healthy individuals, similar to the existence of hematological standards.

The proteomic analysis of tear films in healthy male and female dogs, conducted using MALDI-TOF mass spectrometry, revealed significant differences in the expression of specific proteins, suggesting potential sex-related variations in ocular surface physiology. These findings provide insights into the molecular composition of the canine tear films and their potential regulation by hormonal and physiological factors.

### 4.1. Sex-Related Differences in Tear Film Proteins

Most proteins identified in the tear film exhibited higher expression in females compared to males. Notably, proteins such as metalloproteinase inhibitor 2 (TIMP-2), ATP-dependent 6-phosphofructokinase (PFK), and annexin A13 showed significantly elevated levels in females. These proteins are associated with tissue protection, metabolic activity, and membrane stabilization, respectively. In contrast, interleukin-33 (IL-33), a cytokine involved in immune and inflammatory responses, was more abundant in males.

Similar sex-related differences in protein expression have been reported in other species and biological fluids. For instance, studies have shown that TIMP-2 expression does not exhibit significant sex differences in certain tissues, such as the neocortex of mice [24]. However, in adipose tissue, TIMP-2 mRNA expression was downregulated in high-fat diet-fed male mice but not in females [25]. Regarding PFK, research indicates that male-predominant glycolytic activity is due to the higher expression of Pfkfb3 (phosphofructokinase-2) in males [26]. As for IL-33, sex-determined differences in its expression by innate immune cells have been observed in response to myelin peptide immunization, regulating disease susceptibility in mice [27].

These findings suggest that sex-related differences in the expression of these proteins may not be unique to dogs and could reflect broader physiological patterns across species. However, direct comparisons between tear film compositions in different species remain limited, and further studies are needed to clarify whether these patterns are consistent across mammals.

### 4.2. Comparative Perspective on Sex-Related Tear Film Differences

Rather than focusing solely on hormonal influences, it is valuable to compare the observed sex-related differences in tear film proteins with findings in other species. Previous studies on human tear proteomics have identified sex-related variations in several proteins involved in inflammation, metabolism, and tissue homeostasis. In other mammalian species, sex differences in the expression of TIMP-2, PFK, and IL-33 have been noted in tissues such as adipose tissue, brain, and immune cells [24,26,27].

Further research is needed to determine whether these differences influence ocular health and disease susceptibility in a clinically significant manner.

### 4.3. Implications for Ocular Health

The observed differences in protein expression suggest that tear films in female dogs may provide stronger protection against oxidative stress and tissue damage. This could influence susceptibility to certain ocular surface disorders. For instance, in chronic superficial keratitis (CSK), studies suggest that male dogs are more frequently affected than females, with the lowest risk observed in intact females, while spayed females appear to be at the highest risk [28]. However, for many other canine ocular diseases, there is limited evidence regarding the influence of sex on disease prevalence. Further studies are needed to determine whether the observed sex-related differences in protein expression translate into differential susceptibility to ocular disorders in dogs.

### 4.4. Strengths and Limitations

This study highlights the value of proteomic approaches, such as MALDI-TOF, in elucidating the molecular composition of tear films. However, several limitations should be acknowledged. The small sample size and the focus on healthy dogs restrict the generalizability of the findings. Additionally, potential confounding factors, including variations in breed, age, and environmental conditions, may have influenced the results. Furthermore, the relatively small number of differentially expressed proteins detected limits the robustness of the conclusions. Future studies should address these limitations by incorporating larger, more diverse cohorts and exploring how sex-related differences manifest in diseased states, as well as whether hormonal modulation could serve as a therapeutic strategy.

## 5. Conclusions

The proteomic analysis highlighted significant sex-related differences in tear film composition, likely influenced by hormonal regulation. These findings provide a foundation for further research into the role of tear film proteins in ocular health and disease.

## Figures and Tables

**Figure 1 animals-15-00904-f001:**
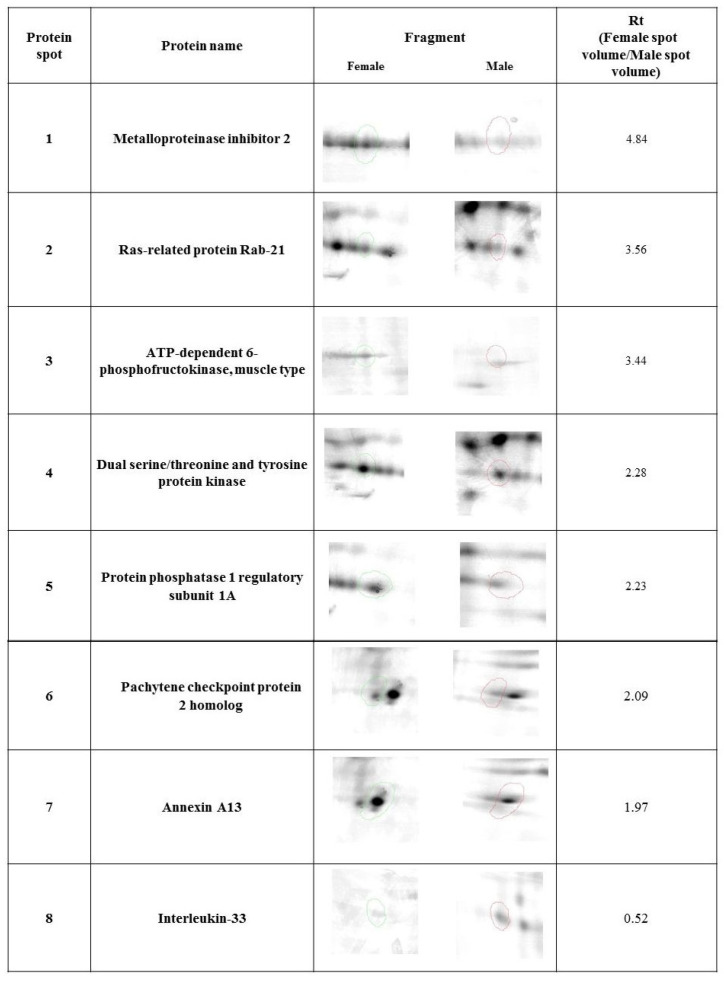
Statistically significant (*p* ≤ 0.05) representatives of 2DE gel spots in the female group compared to the male group, as revealed by the Delta2D software. Protein spots from the female group are marked in green, and protein spots from the male group are marked in red.

**Figure 2 animals-15-00904-f002:**
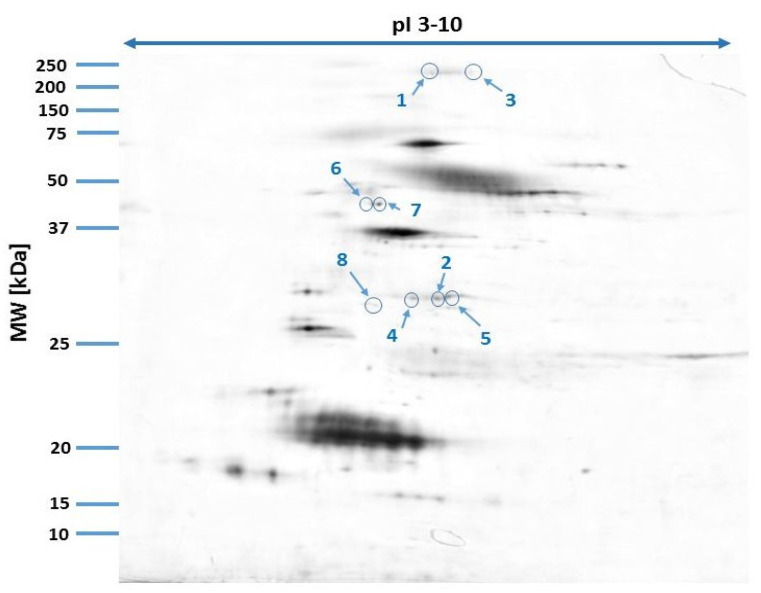
Fused image showing the condensed spot patterns from the experiment. Differentially expressed proteins are marked in blue.

**Figure 3 animals-15-00904-f003:**
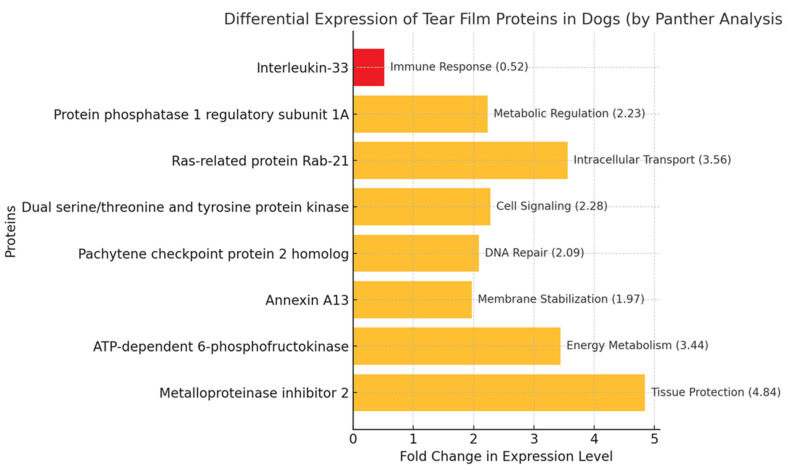
Biological function of proteins identified in the tear films of healthy male and female dogs based on PANTHER program. Proteins with fold changes of <0.67 are considered down-regulated (highlighted in red). Proteins with fold changes of >1.5 are considered up-regulated (highlighted in yellow).

**Table 1 animals-15-00904-t001:** Significantly (*p* ≤ 0.05) differentially expressed proteins in healthy dogs identified by MALDI-TOF MS.

ID	Protein	Accesion Number (UniProtKB)	Score	Match	MI (Da) *	Pi **	Modif.	Seq. Cov (%)	*t*-Test *p*	Species
1	Metalloproteinase inhibitor 2	Q9TTY1	91	6	24,965	8.21	C (C), Ox (M)	30	0.04	*Canis lupus familiaris*
2	ATP-dependent 6-phosphofructokinase, muscle type	P52784	100	10	86,362	8.37	C (C), Ox (M)	6	0.01	*Canis lupus familiaris*
3	Annexin A13	Q29471	70	5	35,628	5.43	C (C), Ox (M)	11	0.03	*Canis lupus familiaris*
4	Pachytene checkpoint protein 2 homolog	E2R222	62	5	48,918	5.83	C (C), Phospho (ST)	7	0.04	*Canis lupus familiaris*
5	Dual serine/threonine and tyrosine protein kinase	Q4VSN4	147	13	106,483	6.25	C (C), Ox (M)	11	0.05	*Canis lupus familiaris*
6	Ras-related protein Rab-21	P55745	84	6	24,545	8.11	C (C), Ox (M)	19	0.02	*Canis lupus familiaris*
7	Protein phosphatase 1 regulatory subunit 1A	Q8WMS3	76	5	19,045	6.09	C (C), Ox (M), Ac (Protein N-term)	14	0.03	*Canis lupus familiaris*
8	Interleukin-33	Q5ZMN3	63	4	30,673	8.45	C (C),	11	0.03	*Canis lupus familiaris*

Abbreviations: C (C)—carbamidomethylation of cysteine; Ox (M)—oxidation of methionine; Ac (protein N-term)—acetylation of protein N-term; Phospho (ST)—phosphorylation of serine or threonine; *—monoisotopic mass; **—calculated pI. Listed molecular weights and pI values correspond to the MASCOT search results.

## Data Availability

The original contributions presented in this study are included in the article Further inquiries can be directed to the corresponding author.

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
