# Peer review of "Proteomic Analysis of Tear Films in Healthy Female and Male Dogs Using MALDI-TOF (Matrix Assisted Laser Desortion/Ionization Time-of-Flight) Mass Spectrometry"

_animals, 2025, doi:10.3390/ani15070904_

Round 1
Reviewer 1 Report
Comments and Suggestions for Authors
Review the methodology in detail, separating the statistical analysis into a single section, since it is confusing for the reader. In the Results section it is mixed with the discussion which is incorrect. The discussion should be more in-depth, since the information from the introduction and results is repeated.

Author Response
|
1. Summary |
|
|
|
Thank you very much for taking the time to review this manuscript. Please find the detailed responses below and the corresponding revisions highlighted in track changes in the re-submitted files.
|
||
|
2. Questions for General Evaluation |
Reviewer’s Evaluation |
Response and Revisions |
|
Does the introduction provide sufficient background and include all relevant references? |
Yes/Can be improved/Must be improved/Not applicable |
|
|
Are all the cited references relevant to the research? |
Yes/Can be improved/Must be improved/Not applicable |
|
|
Is the research design appropriate? |
Yes/Can be improved/Must be improved/Not applicable |
We acknowledge the need for improvements in the research design and have made the necessary modifications |
|
Are the methods adequately described? |
Yes/Can be improved/Must be improved/Not applicable |
The Methodology section has been refined. |
|
Are the results clearly presented? |
Yes/Can be improved/Must be improved/Not applicable |
Result section has been also enhanced. |
|
Are the conclusions supported by the results? |
Yes/Can be improved/Must be improved/Not applicable |
Minor modification have been made to the Conclusion section. |
|
3. Point-by-point response to Comments and Suggestions for Authors |
||
|
Comments 1: Review the methodology in detail, separating the statistical analysis into a single section, since it is confusing for the reader. |
||
|
Response 1: Thank you for your valuable suggestion. We have carefully reviewed the methodology section and have now separated the statistical analysis into a distinct section to improve clarity and readability. These revisions are reflected in the updated manuscript (line 84-173) |
||
|
Comments 2: In the Results section it is mixed with the discussion which is incorrect. |
||
|
Response 2: As suggested, we have revised the Results section to remove any discussion elements. These changes are reflected in the updated manuscript. Comments 3: The discussion should be more in-depth, since the information from the introduction and results is repeated. Response 2: As suggested, we have expanded the Discussion section to provide a more comprehensive interpretation of the findings. We have also minimized redundancy by removing repeated content from the Introduction and Results. The revised version now includes additional references and comparisons with previous studies. (line 226-269) |
||
|
4. Response to Comments on the Quality of English Language |
||
|
Point 1: The English is fine and does not require any improvement. |
||
|
Response 1: |
||
|
5. Additional clarifications |
||
|
In addition to addressing all reviewer comments, we have carefully revised the manuscript to enhance clarity and scientific depth. We have also corrected minor language issues and improved the overall structure where necessary. We believe these revisions have strengthened the manuscript and improved its readability." |
||
Reviewer 2 Report
Comments and Suggestions for Authors
The authors performed 2D gel electrophoresis and MALDI-TOF to characterize the proteome in tear films of healthy male (n=11) and female (n=11) dogs. The same group of authors published a number of methodologically similar proteomic studies that mainly differs in the selection of study subjects and sample types. The authors identified sex-specific expression in 8 out of 446 protein spots. The manuscript contains greatly exaggerated claims about the significance and potential impact of these data. Exaggerated claims must be tuned down before this manuscript can be published. The experimental design is not sufficiently clearly documented and may be inappropriate. I find the ethics statement in its current form inacceptable.
Specific comments:
(1) Were any attempts made to identify the proteins in the 438 spots that did not show differential expression? If yes, this information should be added as supplementary data.
(2) The manuscript contains by far too many unsubstantiated speculations and exaggerated claims. This must be thoroughly revised and the authors are encouraged to decisively shorten their manuscript and focus on objective data. Examples include (list is not comprehensive):
a) lines 19-25. All of this is over-interpretation or speculation. Either delete or rephrase much more cautiously.
b) lines 32-34 & 36/37. Same as above. With 8 differentially expressed proteins that were averaged across groups of 11 male and 11 female dogs, I really don't see a direct relevance for the development of "personalized treatments".
c) lines 69-73. There is no reference for this far-reaching claim. To me, this reads like wishful thinking of the authors to advertise "their" methodology.
(3) lines 55-57. This sentence needs a reference. The introduction should at least briefly summarize the state of the art in human ophthalmology and human tear proteomic analysis. Are there any established clinical applications in human ophthalmology or are such analyses in humans also purely restricted to basic research?
(4) line 69: The citation is incorrectly formatted and the reference is missing from the list of references at the end.
(5) lines 81/82: ...in healthy male and female dogs (plural)
(6) line 83: "Potential differences in tear protein composition between the sexes might help explain a better understanding of the mechanisms that predispose certain breeds to certain diseases." This statement is incomprehensible to me. If breed-specific differences are of interest, then a different experimental design is required! This sentence must be deleted.
(7) lines 83/84: Is the collection of tear fluid with Schirmer strips really non-invasive? I find this procedure performed on myself and my own eyes highly unpleasant and far more distressing than e.g. a simple blood draw.
(8) lines 90-91: The numbers don't add up to 22. A supplementary table on each sampled dog needs to be added with more details on all performed diagnostic tests and their results. I consider it a MAJOR limitation of the study that the male and female group were not breed-matched. How do yo know that the detected differences in protein expression were not caused by different breed rather than by different sex? This limitation and the confounding influence of breed must be transparently stated in the discussion.
(9) Were all 22 dogs sexually intact? This must be clearly stated. If the 22 dogs comprised intact and neutered dogs the confounding factors become inacceptable and then this analysis should not be published.
(10) lines 99-102: I am not familiar with Polish law, but I assume that it must be in line with EU legislation. The study was performed on healthy dogs. Therefore, there was no medical reason to perfom any diagnostics on them. In my country, the performed examination would very clearly constitute an animal experiment that requires some sort of authorization. The authors state that blood serum biochemistry was analyzed. How is this possible without an invasive procedure, i.e. taking a blood sample?
(11) lines 174-176: This is from the manuscript template. The authors must adhere to this, but the text should not appear in the final manuscript.
(12) Figure 3: Delete this figure and the legend. This overly simplistic functional categorisation of 8 proteins is not helpful or informative for the reader.
(13) Lines 211-267. This entire section should be deleted. The statements are overly simplistic and some statements from the literature are out of context and completely misleading.
(14) Line 282: correct --> physiologic
(15) Lines 289-313: This part of the discussion is overly speculative and the cited literature again is taken out of context and not appropriate to support the claims. I suggest to massively shorten the speculations about hormonal influences. Instead, the authors should try to summarize and discuss what has been found in other species. Was any of the 8 identified proteins shown before to be differentially epxressed in males and females? In another species? In humans? In another organ or sample type?
(16) Lines 314-321: Is anything known about sex-specific prevalences of ocular diseases in dogs? If yes, then this needs to be stated with supporting references. If nothing is documented in the scientific literature about the influence of sex on the risk for specific diseases, the entire paragraph should be deleted.
(17) Lines 322-327: It is commendable that the authors discuss limitations of their study. However, the most obvious and important limitations are not mentioned. This needs to be thoroughly revised. Add statements on confounding factors such as different breeds, different ages, non-standardized environment, small number of detected differnetially expressed proteins.
(18) Lines 332-333: "...with potential implications for personalized veterinary ophthalmology and therapeutic interventions." Delete this last phrase. This is again much to far-fetched and speculative.
(19) Line 341: Again I do not agree with the authors: The dogs in this study were subjected to blood draws, tear collection with Schirmer strips, intraocular pressure measurements, fundus evaluations and other diagnostic tests. These tests are anvasive or may be assumed to cause discomfort to the study dogs.
Author Response
1. Summary
Thank you very much for taking the time to review this manuscript. Please find the detailed responses below and the corresponding revisions/corrections highlighted in track changes in the re-submitted files.
2. Point-by-point response to Comments and Suggestions for Authors
Comments 1: The authors performed 2D gel electrophoresis and MALDI-TOF to characterize the proteome in tear films of healthy male (n=11) and female (n=11) dogs. The same group of authors published a number of methodologically similar proteomic studies that mainly differs in the selection of study subjects and sample types. The authors identified sex-specific expression in 8 out of 446 protein spots. The manuscript contains greatly exaggerated claims about the significance and potential impact of these data. Exaggerated claims must be tuned down before this manuscript can be published. The experimental design is not sufficiently clearly documented and may be inappropriate. I find the ethics statement in its current form inacceptable.
Specific comments:
(1) Were any attempts made to identify the proteins in the 438 spots that did not show differential expression? If yes, this information should be added as supplementary data.
Response 1: Thank you for your detailed feedback. We appreciate your critical assessment and have made the necessary revisions to address your concerns.
We have carefully revised the manuscript to ensure that our conclusions are more measured and aligned with the study's findings. We have toned down any overstated claims and provided a more balanced discussion of the results. Additionally, we have clarified the limitations of the study to reflect the exploratory nature of the findings more accurately.
We did not attempt to identify the proteins in the 438 spots that did not show statistically significant differences in expression. Our study focused on the identification of differentially expressed proteins rather than performing a full proteome characterization.
Comments 2: The manuscript contains by far too many unsubstantiated speculations and exaggerated claims. This must be thoroughly revised and the authors are encouraged to decisively shorten their manuscript and focus on objective data. Examples include (list is not comprehensive):
a) lines 19-25. All of this is over-interpretation or speculation. Either delete or rephrase much more cautiously.
Response 2: We acknowledge the need to reduce speculative statements and have thoroughly revised the manuscript to ensure that all interpretations are based strictly on the obtained data. Overstated claims have been removed or rephrased to provide a more balanced discussion.
Regarding the specific comment on lines 19-25, we have carefully revised this section to eliminate over-interpretation
b) lines 32-34 & 36/37. Same as above. With 8 differentially expressed proteins that were averaged across groups of 11 male and 11 female dogs, I really don't see a direct relevance for the development of "personalized treatments".
Response: We have revised the manuscript to remove overstatements regarding the potential for personalized treatments and have reworded this section to more accurately reflect the scope of our findings . Additionally, we have acknowledged the small sample size as a limitation and have explicitly addressed this in the 'Limitations' section of the manuscript.
c) lines 69-73. There is no reference for this far-reaching claim. To me, this reads like wishful thinking of the authors to advertise "their" methodology.
Response: This sentence has been removed from the manuscript.
(3) lines 55-57. This sentence needs a reference. The introduction should at least briefly summarize the state of the art in human ophthalmology and human tear proteomic analysis. Are there any established clinical applications in human ophthalmology or are such analyses in humans also purely restricted to basic research?
Response: The reference has been added.
We have revised the Introduction to include a brief summary of the current state of human ophthalmology and tear proteomic analysis. We have also addressed the potential clinical applications of tear proteomics in human medicine (lines 56-64).
(4) line 69: The citation is incorrectly formatted and the reference is missing from the list of references at the end.
Response: The sentences with incorrectly formatted reference have been removed.
(5) lines 81/82: ...in healthy male and female dogs (plural)
Response: It has been corrected.
(6) line 83: "Potential differences in tear protein composition between the sexes might help explain a better understanding of the mechanisms that predispose certain breeds to certain diseases." This statement is incomprehensible to me. If breed-specific differences are of interest, then a different experimental design is required! This sentence must be deleted.
Response: This sentence has been deleted.
(7) lines 83/84: Is the collection of tear fluid with Schirmer strips really non-invasive? I find this procedure performed on myself and my own eyes highly unpleasant and far more distressing than e.g. a simple blood draw.
Response: Thank you for your comment. We acknowledge that the perception of discomfort during Schirmer strip testing may vary between individuals. However, according to standard medical definitions, an invasive procedure is typically characterized by the penetration or disruption of body tissues, such as in the case of blood sampling, which requires skin puncture. In contrast, Schirmer strip testing does not break tissue continuity; it involves passive absorption of tear fluid from the ocular surface without causing structural damage. Based on this definition, we consider Schirmer strip testing to be a non-invasive procedure.
(8) lines 90-91: The numbers don't add up to 22. A supplementary table on each sampled dog needs to be added with more details on all performed diagnostic tests and their results. I consider it a MAJOR limitation of the study that the male and female group were not breed-matched. How do yo know that the detected differences in protein expression were not caused by different breed rather than by different sex? This limitation and the confounding influence of breed must be transparently stated in the discussion.
Response: Thank you for your valuable feedback. We have carefully reviewed the numbers and revised in the manuscript. We acknowledge that the male and female groups were not breed-matched, and we recognize that breed differences could influence protein expression. This has now been explicitly stated as a limitation in the Discussion section. While our study aimed to investigate sex-related differences, we understand that breed could act as a confounding factor. Future studies with breed-matched groups will be necessary to further validate our findings.
(9) Were all 22 dogs sexually intact? This must be clearly stated. If the 22 dogs comprised intact and neutered dogs the confounding factors become inacceptable and then this analysis should not be published.
Response: Yes, all of 22 dogs were intact. This information was added in the manuscript.
(10) lines 99-102: I am not familiar with Polish law, but I assume that it must be in line with EU legislation. The study was performed on healthy dogs. Therefore, there was no medical reason to perfom any diagnostics on them. In my country, the performed examination would very clearly constitute an animal experiment that requires some sort of authorization. The authors state that blood serum biochemistry was analyzed. How is this possible without an invasive procedure, i.e. taking a blood sample?
Response: Thank you for your comment. The study was conducted in accordance with Polish law, which aligns with EU legislation. All dogs included in the study were presented by their owners for voluntary preventive health check-ups at the clinic for healthy animals. As part of routine veterinary preventive care, the dogs underwent a clinical examination, which included ophthalmic assessment and blood sampling. All owners provided informed consent for tear collection and the performed diagnostics.
Regarding the ethical classification of the study, Polish regulations distinguish between routine veterinary diagnostic procedures and animal experiments. Since no experimental treatment or intervention beyond standard veterinary care was conducted, ethical approval was not required.
As for the blood serum biochemistry, this was performed using blood samples obtained as part of the routine preventive examinations. The collection of blood samples was not conducted specifically for this study but was part of the standard diagnostic assessment performed at the clinic.
(11) lines 174-176: This is from the manuscript template. The authors must adhere to this, but the text should not appear in the final manuscript.
Response: This statements has been removed.
(12) Figure 3: Delete this figure and the legend. This overly simplistic functional categorisation of 8 proteins is not helpful or informative for the reader.
Response: Figure 3 and the legend have been deleted.
(13) Lines 211-267. This entire section should be deleted. The statements are overly simplistic and some statements from the literature are out of context and completely misleading.
Response: Lines 211-267 have been removed.
(14) Line 282: correct --> physiologic
Response: Thank You, it has been corrected.
(15) Lines 289-313: This part of the discussion is overly speculative and the cited literature again is taken out of context and not appropriate to support the claims. I suggest to massively shorten the speculations about hormonal influences. Instead, the authors should try to summarize and discuss what has been found in other species. Was any of the 8 identified proteins shown before to be differentially epxressed in males and females? In another species? In humans? In another organ or sample type?
Response: Thank you for your insightful comments. We have carefully revised this section of the discussion to reduce speculative statements and ensure that all interpretations are well-supported by appropriate references. The discussion on hormonal influences has been significantly shortened, and we have instead focused on summarizing findings from other species.
Additionally, we have reviewed the literature to determine whether any of the eight differentially expressed proteins identified in our study have previously been reported as sex-dependent in other species, including humans and different tissues or biological fluids. This information has now been incorporated into the revised discussion
(16) Lines 314-321: Is anything known about sex-specific prevalences of ocular diseases in dogs? If yes, then this needs to be stated with supporting references. If nothing is documented in the scientific literature about the influence of sex on the risk for specific diseases, the entire paragraph should be deleted.
Response: Thank you for your comment. We have carefully reviewed the available literature regarding sex-specific prevalence of ocular diseases in dogs. Some studies suggest that certain ocular conditions, such as chronic superficial keratitis, may have a sex-related predisposition. We have now incorporated this information into the revised manuscript and provided appropriate references to support these claims.
However, we acknowledge that for many other ocular diseases, there is limited or inconclusive evidence on sex-related differences. Where no supporting literature was available, we have removed speculative statements to ensure that the discussion remains well-grounded in existing scientific knowledge.
(17) Lines 322-327: It is commendable that the authors discuss limitations of their study. However, the most obvious and important limitations are not mentioned. This needs to be thoroughly revised. Add statements on confounding factors such as different breeds, different ages, non-standardized environment, small number of detected differnetially expressed proteins.
Response: Thank you for your valuable feedback. We have thoroughly revised the 'Limitations' section to ensure that all relevant confounding factors are transparently discussed. The revised manuscript now explicitly acknowledges the influence of different breeds, variations in age, non-standardized environmental conditions, and the small number of differentially expressed proteins detected.
(18) Lines 332-333: "...with potential implications for personalized veterinary ophthalmology and therapeutic interventions." Delete this last phrase. This is again much to far-fetched and speculative.
Response: Last phrase has been deleted.
(19) Line 341: Again I do not agree with the authors: The dogs in this study were subjected to blood draws, tear collection with Schirmer strips, intraocular pressure measurements, fundus evaluations and other diagnostic tests. These tests are anvasive or may be assumed to cause discomfort to the study dogs.
Response: We have clarified this issue in the comments above.
4. Response to Comments on the Quality of English Language
Point 1: (x) The English is fine and does not require any improvement.
Response 1:
5. Additional clarifications
In addition to addressing all reviewer comments, we have carefully revised the manuscript to enhance clarity, accuracy, and scientific depth.
We sincerely appreciate the constructive feedback from the reviewers, which has helped us improve the manuscript. We believe that the revised version is now stronger and better aligned with the journal's standards. We look forward to further consideration of our work.
Round 2
Reviewer 2 Report
Comments and Suggestions for Authors
The manuscipt has greatly improved and I appreciate how well the authors responded to my initial comments. I am now happy with the content. Some minimal typographical errors should be corrected during the production stage of the manuscript:
line 72: males --> male
line 79: intact --> sexually intact
Figures need to be renumbered (as figure 3 was deleted)
Author Response
Thank you for your review. All suggested corrections have been implemented in the manuscript.